# Design of Composites by Infiltration Process: A Case Study of Liquid Ir-Si Alloy/SiC Systems

**DOI:** 10.3390/ma14206024

**Published:** 2021-10-13

**Authors:** Rada Novakovic, Simona Delsante, Donatella Giuranno

**Affiliations:** 1Institute of Condensed Matter Chemistry and Technologies for Energy, National Research Council of Italy (ICMATE-CNR), Via De Marini 6, 16149 Genoa, Italy; simona.delsante@unige.it (S.D.); donatella.giuranno@ge.icmate.cnr.it (D.G.); 2Department of Chemistry and Industrial Chemistry, Genoa University and Genoa Research Unit of INSTM, Via Dodecaneso 31, 16146 Genoa, Italy

**Keywords:** Ir-Si alloys, modelling, surface tension, viscosity, molar volume, infiltration, SiC

## Abstract

The design of processing routes involving the presence of the liquid phase is mainly associated with the knowledge of its surface and transport properties. Despite this need, due to experimental difficulties related to high temperature measurements of metallic melts, for many alloy systems neither thermodynamic nor thermophysical properties data are available. A good example of a system lacking these datasets is the Ir-Si system, although over the last fifty years, the structures and properties of its solid phases have been widely investigated. To compensate the missing data, the Gibbs free energy of mixing of the Ir-Si liquid phase was calculated combining the model predicted values for the enthalpy and entropy of mixing using Miedema’s model and the free volume theory, respectively. Subsequently, in the framework of statistical mechanics and thermodynamics, the surface properties were calculated using the quasi-chemical approximation (QCA) for the regular solution, while to obtain the viscosity, the Moelwyn-Hughes (MH) and Terzieff models were applied. Subsequently, the predicted values of the abovementioned thermophysical properties were used to model the non-reactive infiltration isotherm of Ir-Si (eutectic)/SiC system.

## 1. Introduction

Since the late 1970s, iridium silicides have become attractive for applications in electronic industry [1,2,3,4], and much research has been dedicated to thin film deposition, nucleation and growth of Ir-Si intermetallic phases formed and their stabilities have contributed to optimization of electronic devices [5,6]. Some of Ir-Si intermetallic compounds, such as IrSi, $Ir_{3}Si_{5}$ [7] and $IrSi_{3}$ [4,7,8] are of interest for microelectronic or thermoelectric applications. Indeed, among Ir-Si intermetallic compounds, $Ir_{3}Si_{5}$ is a wide-gap semiconductor, while when doped with another transition metal, it may be suitable for use at high temperatures [1,4,7]. Other potential applications of metallic silicides are their use as structural materials in aggressive environments [9] or as infiltrants in the manufacturing of composites via infiltration [10,11,12,13]. Binary Ir-X (X = Ti, Ta, Nb, Hf, Zr, and V) and Ir-Nb-Si ternary refractory superalloys provide a wider operating temperature range, i.e., up to 300 K higher with respect to that of Ni-based alloys as well as better mechanical properties; therefore, these alloys can be used for ultra-high-temperature applications [14,15]. Thermodynamic data of the Ir-Si system are scarce, while for its liquid alloys, due to the experimental difficulties related to high temperature measurements, reactivity of alloy melts with container materials, and oxygen, the complete lack of data can be observed [16]. Therefore, until now, the Ir-Si phase diagram has not been well assessed [17]. Concerning the solid-state measurements, several structural investigations of Ir-Si intermetallic phases by X-ray experiments have been carried out [1,5,18,19,20,21,22,23]. Indeed, $IrSi_{3}$ [1,3,7,19,20,21,24,25], IrSi [1,7,18,19,20,24,25,26,27], $IrSi_{5}$ [20] $Ir_{3}Si_{4}$ [1,2,20,25], $Ir_{4}Si_{5}$ [1,20,25] $Ir_{3}Si_{1.5}$ [20] and $Ir_{3}Si_{5}$ [1,7,23,25] and their structures have been investigated. In the case of well-defined stoichiometry, the corresponding structural datasets differ within experimental errors. Most of the work involving iridium silicides deals with deposition, the phase formation, growth and stability of nonequilibrium thin films and their thermophysical properties in the solid state [1,6,7]. Searcy and Finnie have investigated the thermodynamics of platinum metal silicides with carbon and performing the studies on the phase stability of the Ir-Si-C system at T = 1613 K, the heat of formation of seven iridium silicides has been evaluated [24]. SiC and iridium silicides have been proven to be thermodynamically stable and have good high-temperature oxidation resistance comparable to that of $MoSi_{2}$ [9,24]. Standard enthalpy of formation of IrSi has been determined by high-temperature mixing calorimetry [26,27,28]. The Si-rich side (50–100 at % Si) of the Ir-Si system was investigated by means of density, differential thermal analysis (DTA), X-ray powder diffraction, metallography, microprobe analysis and electrical resistivity [1], and the results obtained were used for the Ir-Si phase diagram assessment [16]. Further investigations of the Ir-Si system were carried out by XRD, EPMA and SEM analyses and are related to the phases and microstructural evolution of Ir-rich alloys (0–50 at % Si) [25]. Based on the abovementioned experimental datasets, the last compilation of Ir-Si thermodynamic data resulting in the most recent Ir-Si phase diagram [17] indicates the presence of $Ir_{3}Si$, $Ir_{2}Si$, $Ir_{3}Si_{2}$, IrSi, $Ir_{4}Si_{5}$, $Ir_{3}Si_{4}$, $Ir_{3}Si_{5}$, $\beta IrSi_{3}$ and $\gamma IrSi_{3}$ intermetallic compounds.

The mixing behaviour of Ir-Si melts can be deduced from the Hume-Rothery empirical factors, such as the size effect ($V_{Ir}/V_{Si}$= 0.861 or its reciprocal value 1.16) [29], valency difference (=0 or 1) and electronegativity difference according to Pauling (=0.3) [30], estimating the driving force for the formation of intermetallic compounds in this system [31]. To compensate for the missing thermodynamic data on the liquid Ir-Si phase, Miedema’s semi-empirical model [32,33,34,35] was combined with the free volume theory [36,37,38,39] in order to predict the enthalpy of mixing and the excess entropy of mixing, respectively. The model-predicted values of these quantities made it possible to evaluate the excess Gibbs free energy and activities of liquid Ir-Si alloys and, thus, to calculate their surface properties using only the models characterised by one interaction energy parameter, such as the quasi-chemical approximation (QCA) for regular solution and/or Butler’s model [40]. However, as in the case of the Co-Si [10,41], the Compound Formation Model (CFM) with four interaction energy parameters would be the most appropriate, but the missing experimental data on the mixing in the Ir-Si liquid phase make its application impossible. Similarly, the calculated values of enthalpy of mixing were used to obtain the viscosity of Ir-Si melts by the Moelwyn-Hughes (MH) model [42] as well as by its extended version with hard sphere contributions [29]. Similar behaviour of the Co-Si and Ir-Si systems in terms of the mixing properties and the corresponding phase diagrams indicates that the thermophysical and structural properties of liquid Ir-Si alloys are very close to those of the Co-Si, described in detail in [41].

*Si-based alloy/SiC* type composites are mainly produced by the infiltration processes [9,10,11,12]. Although this processing route has been in use for a long time, the industries are interested only in the targeting properties to ensure high performances of the final products. The best way to satisfy these needs are material design technologies and applications [10,11,41] that involve the knowledge of the thermophysical and wetting properties of metallic and ceramic components including interactions between them. During the infiltration, the control of reaction is important because, in some cases, the reaction products may cause the degradation of composite mechanical properties and thus, a low reactivity is required [11]. Among the thermophysical properties, the surface tension, viscosity and mass density of alloy melts together with the contact angle between the liquid and solid phases are the key properties to estimate the operating parameters, relevant for the design and optimization of infiltration processes as well as for the simulation of microstructural evolution during solidification. Therefore, taking into account the above-mentioned model-predicted thermophysical properties data, a case study of Si-Ir (eutectic)/SiC system describing the infiltration mechanism and the effects of various factors on it was presented.

## 2. Theory

### 2.1. Predictions of Thermodynamic Properties of Metallic Melts

Thermodynamic modelling is the easiest way to calculate the thermodynamic and subsequently the thermophysical properties of alloy melts in the framework of different models [29,40,41]. However, for many systems, in particular those characterized by high melting temperatures of their alloys, such as the Ir-Si, it is often difficult or even impossible to perform property measurements due to a container and surrounding atmosphere concomitant reactivity to alloy melts. In the present work, the enthalpy of mixing and the excess entropy of Ir-Si melts have been calculated by combining Miedema’s model and free volume theory [32,35,36,37,38], and the results obtained were used to predict the thermophysical properties of this system. In the following, only the basic equations of both formalisms were reported.

After preliminary data analysis of more than 2500 alloy systems, Takeuchi and Inoue [43] have extended Miedema’s model [32,35], introducing a relatively simple description for the enthalpy of mixing $\Delta H_{mix}$ of binary alloy melts. Molar volume $V_{Alloy}$, as a part of $\Delta H_{mix}$, is given as
(1)$$V_{Alloy}=\overset{2}{\underset{\begin{array}{c} i,j=1\\ i\neq j \end{array}}{\sum }}\;x_{i}\times {\left\{\left[1+a\times \left(1-\frac{x_{i}V_{i}^{2/3}\;}{x_{i}V_{i}^{2/3}+x_{j}V_{j}^{2/3}}\right)\times \left(\varphi_{i}-\varphi_{j}\right)\right]V_{i}^{2/3}\right\}}^{3/2}$$
where $V_{i}$, $V_{j}$, $x_{i}$, $x_{j}$, $\varphi_{i}$ and $\varphi_{j}$ are the molar volumes, compositions and work functions of A and B alloy components and a is a constant. According to the model formalism [43], recalling some equations from model theory and after some algebra, the mixing enthalpy $\Delta H_{mix}$ can be calculated using a cubic polynomial of compositions with respect to both alloy components and an interaction parameter Q evaluated at the equiatomic composition, as follows
(2)$$\Delta H_{mix}=4\left(\overset{3}{\underset{k=0}{\sum }}Q_{k}{\left(c_{A}-c_{B}\right)}^{k}\right)x_{A}x_{B}$$
where $x_{A}$ and $x_{B}$ are the compositions of A and B components of an A−B alloy. In the framework of the free volume theory [36,37,38], the grand partition function for an A−B liquid alloy can be evaluated based on those of pure *A* and *B* liquid metals. Subsequently, defining the Gibbs energy of *A* and *B* pure liquid components, the Gibbs energy of mixing of an A−B alloy can be calculated. Considering the enthalpy of mixing of an alloy in terms of the Gibbs-Helmholtz relation, from the first approximation of the regular solution model, two non-linear equations have been assessed in the form
(3)$$P={\left\{1-4x_{A}x_{B}\left[1-exp\left(\frac{\Omega_{AB}}{k_{B}T}\right)\right]\right\}}^{1/2}$$
(4)$$\Delta H_{mix}=\frac{2x_{A}x_{B}\Omega_{AB}}{P+1}$$
where $\Omega_{AB}$ is the exchange energy and P is a parameter related to the nearly random configuration of atoms in an A−B alloy that can be set approximately to be 1. Once Ω_*AB*_ and *P* are obtained, using the mathematical formalism of the model, including the thermodynamic relations [37]
(5)$$\Delta H_{mix}=\Omega_{AB}x_{A}x_{B}\left(1-x_{A}x_{B}\frac{\Omega_{AB}}{RT}\right)$$
(6)$$\Delta S^{xs}=\Delta S_{CONF}^{xs}+\Delta S_{VIB}^{xs}$$
(7)$$G^{xs}=\Delta H_{mix}-TS^{xs}$$
with $S^{xs}$, $S_{CONF}^{xs}$, $S_{VIB}^{xs}$ and $G^{xs}$ as the excess entropy, its configurational and vibrational parts, and excess Gibbs energy, respectively. Therefore, the excess entropy and excess Gibbs energy can be evaluated using the values of the enthalpy of mixing that is often measurable thermodynamic function and excess volume together with some thermophysical properties of alloy components. $\gamma_{A}$ and $\gamma_{B}$ activity coefficients of A and B alloy components in A−B solution phases are related to the excess Gibbs energy of mixing $\;G^{xs}$ by the standard thermodynamic relation, as
(8)$$G^{xs}=RT\left[x_{A}\times \ln\gamma_{A}+x_{B}\times \ln\gamma_{B}\right]$$

In the framework of QCA for regular solution [40], the activity coefficients of A and B alloy components are expressed by
(9)$$\gamma_{A}={\left[\frac{\beta -1+2x_{A}}{x_{A}\left(1+\beta \right)}\right]}^{Z/2}\;$$
(10)$$\gamma_{A}={\left[\frac{\beta -1+2x_{B}}{x_{B}\left(1+\beta \right)}\right]}^{Z/2}\;$$
where Z is the coordination number and β is the auxiliary function describing the energetics of the bulk phase.

### 2.2. Structural Information: $S_{cc}\left(0\right)$ and $\alpha_{1}$ Microscopic Functions

The concentration–concentration structure factor in the long wavelength limit $S_{cc}\left(0\right)$ is an important microscopic function in describing the nature of mixing of liquid alloys in terms of chemical ordering and segregation (or phase separation) [41,44]. Due to difficulties in diffraction experiments, the theoretical determination of $S_{cc}\left(0\right)$ is of great importance when the nature of atomic interactions in the melt must be analysed. The mixing behaviour of liquid binary alloys can be deduced from the deviation of $S_{cc}\left(0\right)$ from $S_{cc}\left(0,\;id\right)$. The presence of chemical order is indicated when $S_{cc}\left(0\right)$ < $S_{cc}\left(0,id\right);$ on the contrary, if $S_{cc}\left(0\right)$ > $S_{cc}\left(0,id\right)$, the segregation and demixing in liquid alloys take place. Once the Gibbs energy of mixing $G_{M}$ of the liquid phase is known, $S_{cc}\left(0\right)$ can be expressed either by $G_{M}$ or by the activities $a_{A}$ and $a_{B}$, as
(11)$$S_{cc}\left(0\right)=RT{\left(\frac{\partial^{2}G_{M}}{\partial x_{A}^{2}}\right)}_{T,P,N}^{-1}=x_{B}a_{A}{\left(\frac{\partial a_{A}}{\partial x_{A}}\right)}_{T,P,N}^{-1}=x_{A}a_{B}{\left(\frac{\partial a_{B}}{\partial x_{B}}\right)}_{T,P,N}^{-1}$$

For ideal mixing, the energy parameters become zero and Equation (12) becomes
(12)$$S_{cc}\left(0,id\right)=x_{A}\times x_{B}$$

In order to quantify the degree of order and segregation in the melt, another important microscopic function, known as the Warren-Cowley short-range order parameter $\alpha_{1}$ [41,45] is used. $S_{cc}\left(0\right)$ and $\alpha_{1}$ are related to each other, by
(13)$$\frac{S_{cc}\left(0\right)}{x_{A}\times x_{B}}=\frac{1+\alpha_{1}}{1-\left(Z-1\right)\alpha_{1}}$$

For the equiatomic composition, the chemical short-range order (CSRO) parameter, often denoted as $\alpha_{1}$, is found to be −1≤ $\alpha_{1}\leq 1$. The negative values of this parameter indicate ordering in the melt, and complete ordering is manifested by $\alpha_{1}^{min}=-1$. On the contrary, the positive values of $\alpha_{1}$ indicate segregation, while if $\alpha_{1}^{max}=1$, the phase separation takes place.

### 2.3. Surface Properties

Once the excess Gibbs free energy was obtained in the framework of the free volume theory [37,38], the surface segregation and surface tension of binary liquid alloys can be calculated by the quasi-chemical approximation (QCA) for regular solution by
(14)$$\sigma =\sigma_{A}+\frac{k_{B}T\times \left(2-p\times Z\right)}{2\alpha }ln\left(\frac{x_{A}^{s}}{x_{A}}\right)+\frac{Z\times k_{B}T}{2\alpha }\left[p\times ln\frac{\left(\beta^{s}-1+2x_{A}^{s}\right)\left(1+\beta \right)}{\left(\beta -1+2x_{A}\right)\left(1+\beta^{s}\right)}-q\times ln\frac{\left(\beta -1+2x_{A}\right)}{\left(1+\beta \right)x_{A}}\right]$$
(15)$$\sigma =\sigma_{B}+\frac{k_{B}T\times \left(2-p\times Z\right)}{2\alpha }ln\left(\frac{x_{B}^{s}}{x_{B}}\right)+\frac{Z\times k_{B}T}{2\alpha }\left[p\times ln\frac{\left(\beta^{s}-1+2x_{B}^{s}\right)\left(1+\beta \right)}{\left(\beta -1+2x_{B}\right)\left(1+\beta^{s}\right)}-q\times ln\frac{\left(\beta -1+2x_{B}\right)}{\left(1+\beta \right)x_{B}}\right]$$
where $\sigma_{A}$ and $\sigma_{B}$ are the surface tensions of pure components, α is the mean surface area of an A−B alloy and β (Equations (9) and (10)) and $\beta^{s}$ are auxiliary functions describing the energetics of the bulk and surface phase, respectively. p and q are the surface coordination fractions defined as the fractions of the total number of nearest neighbours of an atom in its own layer and that in the adjoining layer. Therefore, p+q=1. For a close-packed structure, the values of these parameters usually are taken as ½ and ¼, respectively. The QCA model has been detailed in [40].

### 2.4. Transport Properties

Viscosity of Ir-Si alloys was evaluated using the Moelwyn-Hughes (MH) model, which is the simplest one among those reported in [29], as well as by the Terzieff viscosity model [46], representing an extension of the Iida model [47]. The MH viscosity isotherm is described by
(16)$$\eta =\left(x_{1}\eta_{1}+x_{2}\eta_{2}\right)\left(1-2x_{1}x_{2}\times \frac{H_{mix}}{RT}\right)$$
where $\eta_{1}$ and $\eta_{2}$ are the viscosities of pure components and $H_{mix}$ is the mixing enthalpy of the alloys. The viscosity isotherm calculated by the Terzieff model [46,48] combines the energetics of a system in terms of thermodynamics and the hard sphere approach. According to that model, the relative viscosity of an alloy melt is expressed as the ratio of the excess viscosity and additive viscosity, in the form
(17)$$\frac{\eta^{xs}}{\eta_{id}}=\alpha \times \frac{x_{1}x_{2}{\left(\;\bar{\sigma_{1}}-\bar{\sigma_{2}}\right)}^{2}}{x_{1}{\bar{\sigma }}_{1}^{2}-x_{2}{\bar{\sigma }}_{2}^{2}}+\beta \times \left[\left(1+\frac{x_{1}x_{2}{\left(m_{1}^{0.5}-m_{2}^{0.5}\right)}^{2}}{{\left(x_{1}m_{1}^{0.5}-x_{2}m_{2}^{0.5}\right)}^{2}}\right)-1\right]+\gamma \times \frac{2H_{mix}}{RT}+\delta \times \frac{2x_{1}x_{2}\left|V_{1}-V_{2}\right|}{V_{1}+V_{2}}$$
where $H_{mix}$, R, $x_{i}$, $m_{i}$,$\;{\bar{\sigma }}_{i}$, $V_{i}$ (i=A,B) are the enthalpy of mixing, gas constant, atomic fraction, atomic mass, hard sphere diameter and molar volume of a component i, respectively. α, β, γ, and δ are statistical weights obtained by fitting to Equation (17) a large number of experimental viscosity datasets at the equiatomic composition. Based on the standard relation $\eta \left(T\right)=\eta {\left(T\right)}_{id}+\eta {\left(T\right)}^{xs}$, the viscosity of binary melts [46] can be expressed by
(18)$$\eta \left(T\right)=\eta {\left(T\right)}_{id}\left(1+\eta {\left(T\right)}^{xs}\right)$$

### 2.5. Non-Reactive Infiltration: Metal/Metal and Metal/Ceramic Composites

The investigation of reactive infiltration as the most complex part of the infiltration process includes chemical reactions that can be predicted theoretically to some extent [11,12]; therefore, it requires the experimental work. On the contrary, the modelling of non-reactive infiltration is possible if the model-predicted thermophysical properties and/or the corresponding literature data are available. Non-reactive infiltration of liquid alloys in contact with porous metallic or ceramic substrates, as the processing route or the processing step, is widely used to produce metal (Me1)/metal (Me2) and metal/ceramic composites [10,11,12,13]. Indeed, taking into account a miscibility gap that characterises Me1/Me2 monotectic metallic systems, Eremenko and Lesnik have performed combined experimental-theoretical investigation of non-reactive infiltration of the Ag/Ni, Ag/Fe, Pb/Ag, Pb/Ni and Pb/Fe systems [49]. Concerning the infiltration of porous ceramics by liquid metallic materials, usually two steps can be distinguished, i.e., the initial step of non-reactive infiltration process, lasting a very short time, followed by reactive infiltration, as it has been observed in the case of the Ni-Si/C [50] and Si/C systems [51,52]. Non-reactive infiltration of porous metallic or ceramic samples with metallic infiltrants follows the laws of the Washburn [53] and Deryagin [54] theory of capillary infiltration. The wetting of the surface is the key factor, and the infiltration rate, controlled by the viscous flow, depends on the infiltrant composition and its thermophysical properties. The depth of infiltration h varies with time t obeying the Washburn parabolic equation [53] as follows,
(19)$$h^{2}=r_{eff}\times \frac{\sigma_{LG}\times \cos\left(\theta \right)}{2\eta }\times t=Kt$$
where θ, $\sigma_{LG}$, η, $r_{eff}$, and K are the contact angle between the liquid and solid phase, surface tension, viscosity, effective pore radius of the preform and constant.

## 3. Results and Discussion

### 3.1. Thermodynamics of Ir-Si Melts: Miedema’s Model and Free Volume Theory

Based on Miedema’s semi-empirical model [32,35] and its refinement [43], the enthalpy of mixing $\Delta H_{mix}$ of liquid Ir-Si alloys (Figure 1; curve 1) can be calculated by
(20)$$\Delta H_{mix}^{Ir-Si}\;\left(\mathrm{kJ}\cdot {\mathrm{mol}}^{-1}\right)=4x_{Ir}x_{Si}\left[-26.2-0.8\left(x_{Ir}-x_{Si}\right)\right]\;$$

In the framework of free volume theory [37,39], the enthalpy of mixing $\Delta H_{mix}$ [43] of liquid Ir-Si alloys and thermophysical property datasets such as the melting temperature, density and molar volume of alloy components [29,55,56] were used to calculate the excess entropy $\Delta S^{xs}$ (Table 1), subsequently combined in Equation (7) to obtain the excess Gibbs free energy (Figure 1; curve 2). Adding the ideal mixture term to Equation (7), the Gibbs free energy of mixing $\Delta G_{M}$ of Ir-Si melts for T = 1873 K was calculated (Figure 1; curve 3). The curve describing $\Delta H_{mix}$ is symmetric, while those of $-T\Delta S^{xs}$ (Figure 1; curve 4), $\Delta G^{xs}$ and Δ*G_M_* are slightly asymmetric with respect to the equiatomic composition (Figure 1). Due to the lack of experimental data, a comparison with model-predicted values is not possible.

In order to evaluate the energetics of this system, the normalised form GMRT is appropriate to characterise the type of interactions between Ir and Si-constituent atoms and, the model predicted value of −1.76 at $x_{Si}$ = 0.52 is an indicator for a moderately interacting system (Figure 2; curve 1), such as the Si-Co [41], Al-Co [57] and others. The calculated values of Ir and Si activities exhibit strong negative deviation from ideality (Figure 2; curves 2a and 2b), comparable to that observed for liquid Co-Si alloys [41].

The phase diagram shows the existence of nine intermetallic compounds in the Ir-Si system [17], and among them, IrSi is energetically favoured. Schlesinger summarised the literature data on the enthalpy of formation of IrSi [58]. A comparison of these data shows that predicted values vary between −27.2 and −67 kJ [59], and only the last value agrees well with the corresponding experimental value of −63.8 kJ [28], later measured as −64.4 kJ [26] and confirmed in [27].

### 3.2. Density/Molar Volume of Ir-Si Melts

The importance of density is twofold. Firstly, it affects the atomic structure and short-range ordering of metallic melts and, on the other side, it is implicitly contained in the Rayleigh number that characterises fluid flow in all technological processes that include the presence of the liquid phase. Therefore, the density or molar volume of liquid alloys are used as an input for numerical simulations of the above-mentioned processes as well as for material property design. Molar volume isotherm of liquid Ir-Si was calculated (Equation (2)) for T = 1873 K (Figure 3; curve 1). Due to the lack of experimental data, a comparison with model-predicted values is not possible. Thus, for a comparison, the molar volume isotherm of Co-Si melts (Figure 3; curve 2) obtained at the same temperature together with the molar volume data deduced from the experimental data of density measured at T = 1773 K [60], including the corresponding isotherm (Figure 3; curve 3), are shown. The dataset [60] has been obtained at a lower temperature, and, therefore, the experimental and theoretical data are lower with respect to the two isotherms calculated for T = 1873 K. The experimental molar volume isotherm of liquid Co-Si alloys exhibits negative deviation from the ideal mixture, and the same behaviour is expected for liquid Ir-Si. In addition, taking into account higher density of liquid Co with respect to that of Ir, higher molar volume predicted values of Ir-Si melts (Figure 3; curve 1) with respect to those of the Co-Si could be deduced.

### 3.3. Structural Information: Concentration Fluctuations in the Long-Wavelength Limit and Chemical Short-Range Order Parameter in Ir-Si Melts

The ordering phenomena in the Ir-Si liquid phase have been analysed by concentration fluctuations in the long-wavelength limit $S_{cc}\left(0\right)$ and short-range order parameter ($\alpha_{1}$) as functions of bulk compositions. The enthalpy of mixing $\Delta H_{mix}$ predicted by Miedema’s model and the excess mixing entropy $\Delta S^{xs}$ obtained in the framework of the free volume theory (Table 1) were combined by means of fundamental thermodynamic equations, and these data were used to calculate the Gibbs free energy of mixing $\Delta G_{M}$ for T = 1873 K. Subsequently, inserting $\Delta G_{M}$ into Equation (11) and the obtained values of $S_{cc}\left(0\right)$ into Equation (13), the two microscopic functions $S_{cc}\left(0\right)$ and $\alpha_{1}$ describing the nature of mixing and the degree of order of Ir-Si melts are shown in Figure 4.

The *S_cc_*(0) curve exhibits the maximum of 0.07, which differs significantly from the corresponding ideal curve $S_{cc}\left(0,id\right)$, i.e., $S_{cc}\left(0\right)$ < $S_{cc}\left(0,id\right)$, indicating the formation of $A_{\mu }B_{\nu }$ heterocoordinated short-range order elements in alloy melts with *μ* and ν stoichiometric coefficients describing the stoichiometry of *IrSi* energetically favoured intermetallic compound [26,27]. The symmetric $\alpha_{1}$-curve with respect to the equiatomic composition has the minimum value of −0.15 and together with its negative values over the whole composition range (Figure 4) substantiate the compound-forming tendency in the Ir-Si system.

### 3.4. Surface Segregation and Surface Tension of Liquid Ir-Si Alloys: QCA Modelling

Surface segregation and surface tension of liquid Ir-Si alloys were calculated by the QCA for regular solution [40]. Based on the abovementioned theoretical considerations and taking into account [32,33,43], the excess Gibbs free energy of the liquid phase (Table 1) was calculated by Equation (7), and together with the molar volume, structural data [29] as well as the surface tension reference data of liquid Ir [56] and Si [61] alloy components were used as input. Combining Equations (14) and (15), the segregation of Si-atoms to the surface of Ir-Si melts, over the whole composition range was obtained (Figure 5).

The presence of nine intermetallic compounds in the Ir-Si system [17] leads to the short-range ordering phenomena occurring at least near the liquidus line, that reduce the degree of segregation up to 25%, as it was observed in the case of liquid Co-Si alloys [41]. Therefore, in the case of Ir-Si melts, the surface enrichment of Si-atoms calculated by the QCA for regular solution seems to be very high, but the lack of thermodynamic data makes impossible the application of a more appropriate model, such as the Compound Formation Model (CFM), which has a good prediction accuracy. As a rule, in all alloy systems, the atoms of a component with lower surface tension segregate to the surface and an increase in temperature reduces the degree of segregation.

Inserting the calculated values of surface composition either into Equation (14) or Equation (15), the surface tension isotherm of Ir-Si liquid alloys (Figure 6; curve 1) can be obtained. The QCA surface tension isotherm is characterized by positive deviations from that calculated by the perfect solution model (Figure 6; curve 2), confirming the opposite deviations of the surface and thermodynamic property curves with respect to the ideal behaviour [29]. Similar behavior of the surface tension was observed for Co-Si melts [41].

### 3.5. Viscosity

The enthalpy of mixing is a common input in the Moelwyn-Hughes [42] and Terzieff [46] viscosity models, described by Equations (16)–(18), respectively. Therefore, to calculate the viscosity of Ir-Si melts, the predicted values of enthalpy of mixing [43], together with the viscosity reference data of liquid Ir [56] and Si [62], were taken as input in the two models. In the case of Terzieff’s model [46,48,63], the inputs, such as the molar volume of Ir and Si [29] and hard sphere diameter of Ir [64] and Si [65] at the melting temperature, were also used. The viscosity of liquid Ir-Si alloys was calculated for T = 1873 K using Moelwyn-Hughes’s (Figure 7a; curve 1) and Terzieff’s (Figure 7a; curve 2) models, and the two curves exhibited similar trends. Higher viscosity values were obtained by the Terzieff model (Figure 7a; curve 2). Without experimental evidence, it is difficult to deduce which one of the two suggested viscosity isotherms is better. One of the most effective strategies to compare the two property curves is to analyse the trend of a property data related to a similar system characterised experimentally. Thus, the viscosity isotherm of Ir-Si melts was calculated by MH (Figure 7b; curve 1) and compared to the corresponding datasets of liquid Co-Si alloys. Indeed, combining the thermodynamic data of the Si-Co liquid phase [66] with the viscosity reference data of liquid Si [62] and Co [55,67], the viscosity isotherm was calculated for T = 1673 K and compared with experimental data [68] obtained at the same temperature (Figure 7b; curve 2). The two viscosity isotherms are similar and exhibit the same trend with the experimental data [68] as shown in Figure 7b. Therefore, based on the above-mentioned data, one may determine that the Moelwyn-Hughes isotherm (Figure 7a; curve 1) better describes the viscosity of liquid Ir-Si alloy. This is further substantiated by the analysis of non-reactive infiltration (see Section 3.6).

### 3.6. Non-Reactive Infiltration: Case Study of a Liquid Ir-Si Alloy/SiC System

Design of metal/metal and metal/ceramic composites by infiltration processing routes includes the experimental and/or theoretical thermophysical properties data of liquid metallic phase and the experimental data on the wettability between liquid and solid phases. Such datasets include the melting temperature, surface tension, density/molar volume and viscosity of liquid metals or alloys, while the wetting is described in terms of the interfacial reactions, contact angle and work of adhesion, together with well-defined operating conditions (working atmosphere, operating temperature, time, type of the substrate, including its porosity, and grain size) [10,41]. The non-reactive infiltration is governed by the viscous flow, and it is related to non-reactive wetting [49,69]. Therefore, the modelling of non-reactive infiltration of Ir-Si/SiC systems is based on the parabolic Washburn’s equation that is used to calculate the infiltration depth (Equation (19)). To this aim, the model predicted thermophysical property datasets, i.e., the surface tension (Equations (14) and (15)) and viscosity (Equations (16)–(18)) of the Ir-80.5Si (at %) eutectic and ${\mathrm{IrSi}}_{3}$ alloys were combined with the corresponding experimental contact angle values of θ = 10° [12] measured at T = 1623 K on SiC substrate with an average pore size (r = 65 μm) [12]. For a comparison, the same type of datasets for the Co-77.9Si (at %) eutectic alloy and pure liquid Si on SiC with the contact angle data of θ = 15° [10] and θ = 15° [70,71], obtained at 1723 and 1853 K, respectively, were used as inputs in Washburn’s equation. The calculated curves are shown in Figure 8.

The infiltration depth in the Ir-80.5Si/SiC system was calculated by Equation (19) using the Moelwyn-Hughes (Figure 8; curve 1a) and the Terzieff (Figure 8; curve 1b) models for prediction of the viscosity. Comparing the two infiltration curves with that of the Co-77.9Si/SiC system (Figure 8; curve 1a), obtained by the Moelwyn-Hughes model, seems to be more appropriate, confirming the viscosity analysis in Section 3.5. The lowest viscosity of liquid Si [62] results in the highest rate of infiltration and thus, the maximum values of the infiltration depth (Figure 8; curve 3).

Design of Si-based alloy/ceramic composites includes the porosity of ceramic preforms with cylindrical pores of radius varying between 0.14 and 1.15 microns [12,50,51,52,72]. The curves that characterised non-reactive infiltration of liquid Ir-80.5Si (at %) into solid SiC-preforms with porosity of 0.65, 0.85 and 1.15 microns are shown in Figure 9. As expected, the infiltration depth (Equation (19)) is proportional to the pore size of a porous SiC-preform and thus, for size (r = 1.15 μm), the maximum values can be observed (Figure 9).

Investigating Si and Si-based alloys in contact with various carbon preforms, many authors have considered non-reactive infiltration as an initial step of reactive infiltration processes [10,11,12,13,50,51,52]. The wetting of carbon by liquid Si and its alloys, followed by the reaction between Si and C, lead to the formation of SiC by the so-called *reaction-bonded silicon carbide* process [9,12,50]. Recent investigation of reactive infiltration of liquid eutectic Ir-80.5Si (at %) alloy into SiC-C porous preforms [12] showed linear infiltration curves (Figure 10a,b) indicating that the process is governed by the reaction accompanied by the change in the melt composition, i.e., from the Ir-80.5Si (at %) eutectic alloy to the $IrSi_{3}$, resulting in an increase in the viscosity. At the end of the infiltration experiments [12], the infiltration curves (Figure 10a,b; curve 1) exhibit a small deviation of linearity that can be explained by a decreased Si-concentration in the alloy with respect to that necessary for SiC formation [52]. Analysing the experimental results related to liquid $IrSi_{3}$/SiC [12], an irregular infiltration front was found, and this may suggest a concomitant role of capillarity and viscous forces, at least at the final stage of reactive infiltration. Therefore, the last part of the infiltration curves may be considered to obey non-reactive infiltration law (Figure 10a,b; curve 2).

Model predicted values of the surface tension and viscosity of liquid $IrSi_{3}$ alloy together with the contact angle values of liquid $IrSi_{3}$/SiC systems measured at T = 1523 and T = 1623 K [12] were used to calculate non-reactive infiltration curves for the final stage of infiltration. Due to a very short duration of this stage, the infiltration curve is only slightly concave and resembles the linear one (Figure 10a,b). In order to have a clear picture of phenomena occurring in the last stage of reactive infiltration, further investigation is needed.

## 4. Conclusions

A case study of liquid Ir-Si alloys/SiC systems is an example of how the lack of experimental thermodynamic, surface and transport properties data, often necessary for material and/or process design, can be compensated by the model-predicted values and subsequently combined with the experimental contact angle data to describe non-reactive infiltration. Therefore, in the present work, a step-by-step guide to property modelling started from the predictions of the enthalpy of mixing and molar volume in terms of Miedema’s model and their subsequent use as input to calculate other thermodynamic functions of mixing in the framework of the free volume theory. Once the energetic part of liquid Ir-Si alloys was defined, those data were used to simulate their thermophysical properties. Indeed, the surface segregation and surface tension of Ir-Si melts were obtained by the quasi-chemical approximation (QCA) for regular solution model, while the Moelwyn-Hughes model and more complex Terzieff’s model were used to predict the viscosity. If possible, it is recommended to compare and discuss the behaviour or trend of model-predicted property values of the newly investigated system with the corresponding data of similar systems available in the literature. Accordingly, a comparison of the present results with the available properties data of Co-Si melts was performed, indicating the Moelwyn-Hughes model as more appropriate to describe the viscosity of Ir-Si melts. Based on the above-mentioned data, the parabolic Washburn’s equation was used to design non-reactive infiltration of Ir-Si/SiC systems in terms of the infiltration depth during the process for different temperatures, time and pore size of SiC preforms.

## Figures and Tables

**Figure 1 materials-14-06024-f001:**
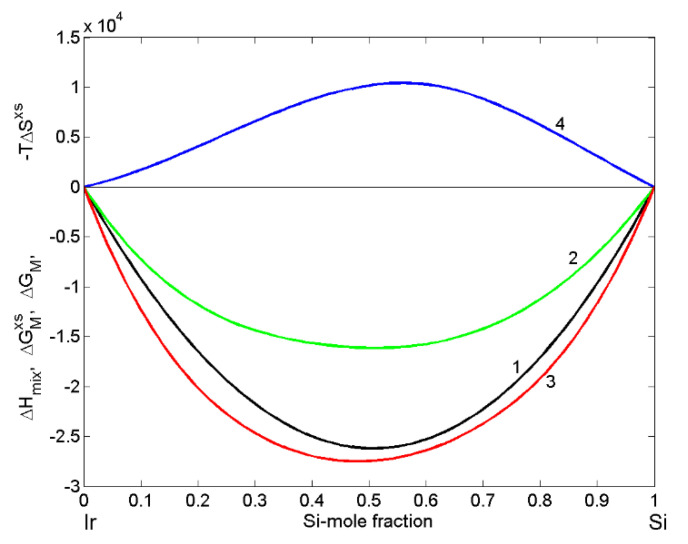
Concentration dependence of thermodynamic properties of liquid Ir-Si alloys calculated for T = 1873 K: 1-the enthalpy of mixing ($\Delta H_{M}$); 2-the excess entropic term ($-T\Delta S^{xs}$); 3-the excess Gibbs free energy ($\Delta G_{M}^{xs}$); 4-the Gibbs free energy of mixing ($\Delta G_{M}$).

**Figure 2 materials-14-06024-f002:**
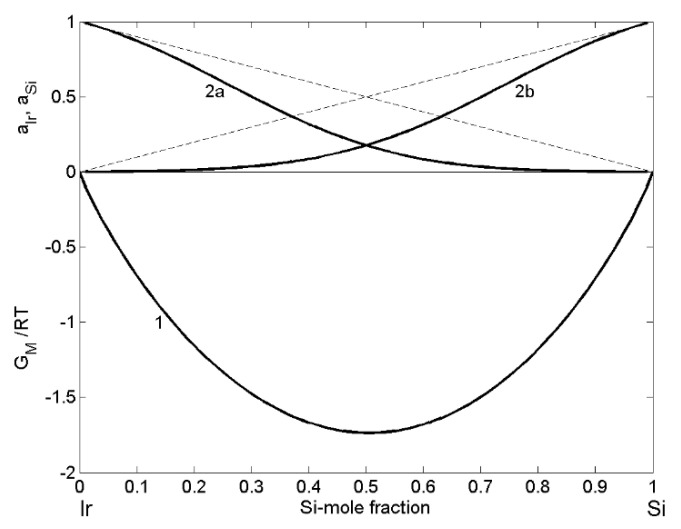
Concentration dependence of the Gibbs free energy of mixing (GMRT; curve 1), the activities of iridium ($a_{Ir}$; curve 2a) and silicon ($a_{Si}$; curve 2b) of liquid Ir-Si alloys calculated for T = 1873 K. (----- the ideal mixture).

**Figure 3 materials-14-06024-f003:**
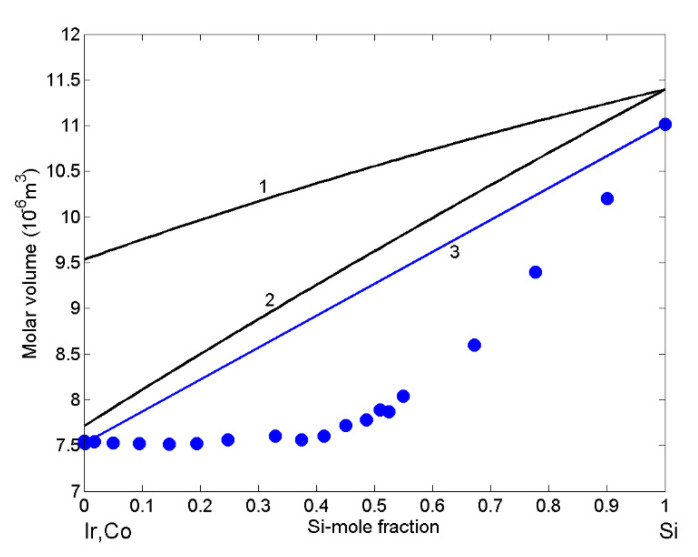
Concentration dependence of the molar volume of liquid Ir-Si (curve 1) and Co-Si (curve 2) alloys calculated for T = 1873 K. For a comparison, the molar volume of Co-Si melts (curve 3) calculated for T = 1773 K together with experimental values (⬤) obtained from the density experimental data [60] measured at the same temperature are shown.

**Figure 4 materials-14-06024-f004:**
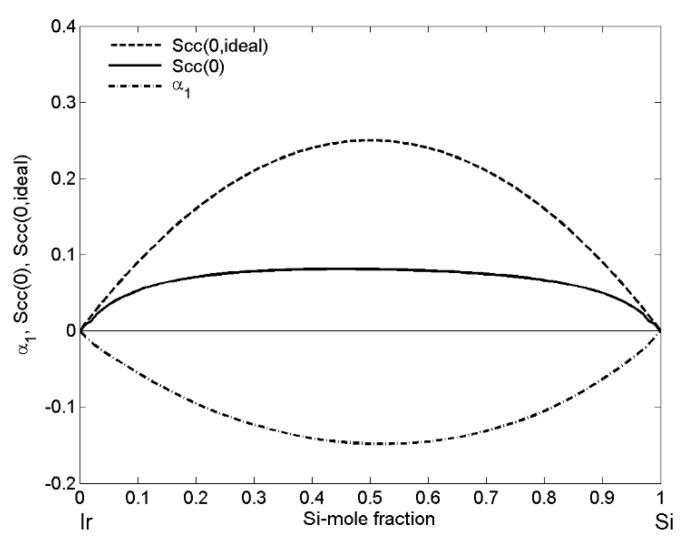
Concentration fluctuations in the long-wavelength limit ($S_{cc}\left(0\right)$ and $S_{cc}\left(0,id\right))$ and chemical short-range order parameter ($\alpha_{1}$) vs. bulk composition ($x_{Si}$) of liquid Ir-Si alloys calculated for T = 1873 K.

**Figure 5 materials-14-06024-f005:**
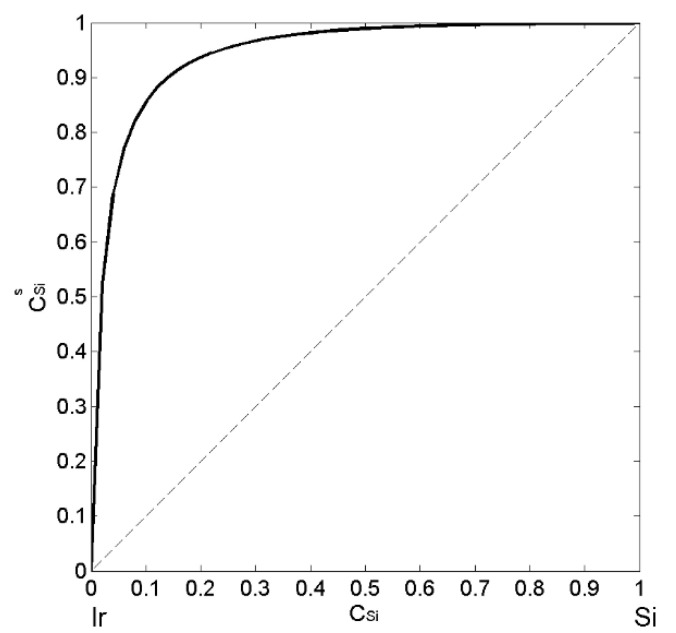
Surface composition ($C_{Si}^{s}$) vs. bulk composition ($C_{Si}$) in liquid Ir-Si alloys calculated by the QCA for regular solution for T = 1873 K. (----- the additive rule).

**Figure 6 materials-14-06024-f006:**
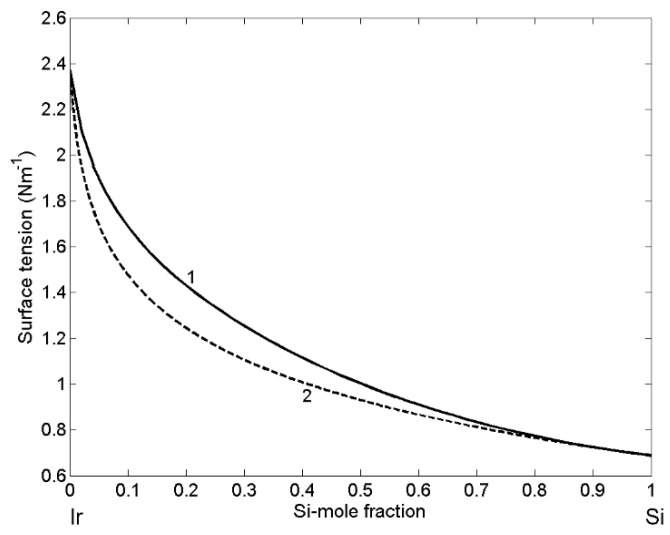
Surface tension isotherm of liquid Ir-Si alloys calculated for T = 1873 K by the QCA for regular solutions. (------) the perfect solution model.

**Figure 7 materials-14-06024-f007:**
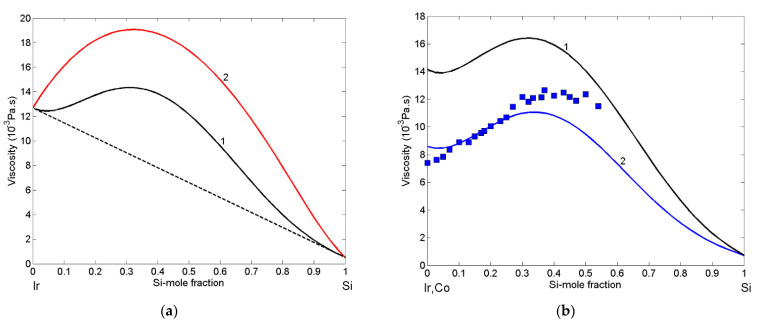
(**a**). Viscosity isotherms of liquid Ir-Si alloys for T = 1873 K calculated by Moelwyn-Hughes’s model (curve 1) and Terzieff’s model (curve 2). (------) the additive rule. (**b**) Viscosity isotherms of liquid Ir-Si (curve 1) and Co-Si (curve 2) alloys calculated by Moelwyn-Hughes’s (MH) model for T = 1773 K. For a comparison, the Co-Si viscosity isotherm (curve 2) and datasets (■) [68] are shown.

**Figure 8 materials-14-06024-f008:**
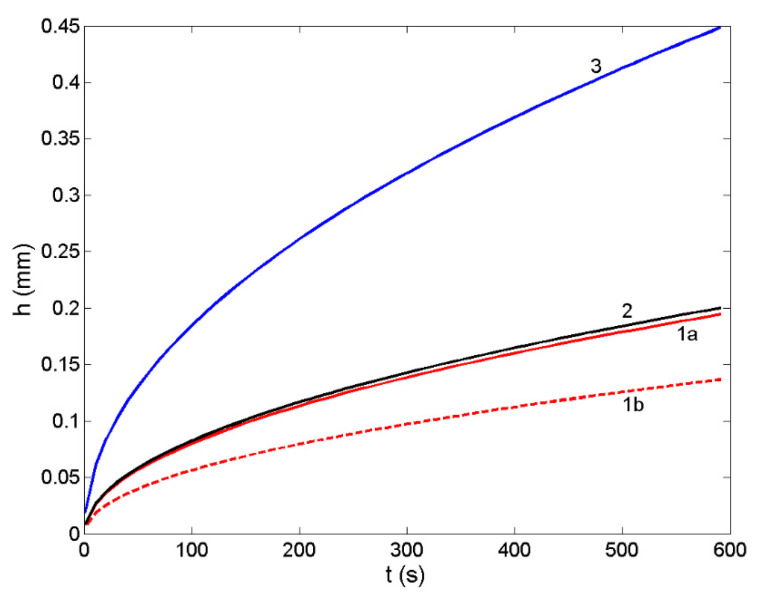
Infiltration depth of liquid Ir-80.5Si (at %) eutectic alloy into a porous SiC-substrate with a pore size of r = 65 μm, calculated for T = 1873 K using different viscosity models: The Moelwyn-Hughes (curve 1a) and Terzieff’s model (curve 1b). For comparison, non-reactive infiltration of liquid Co-77.9Si (at %) eutectic alloy (curve 2) and pure Si (curve 3) on SiC are also shown.

**Figure 9 materials-14-06024-f009:**
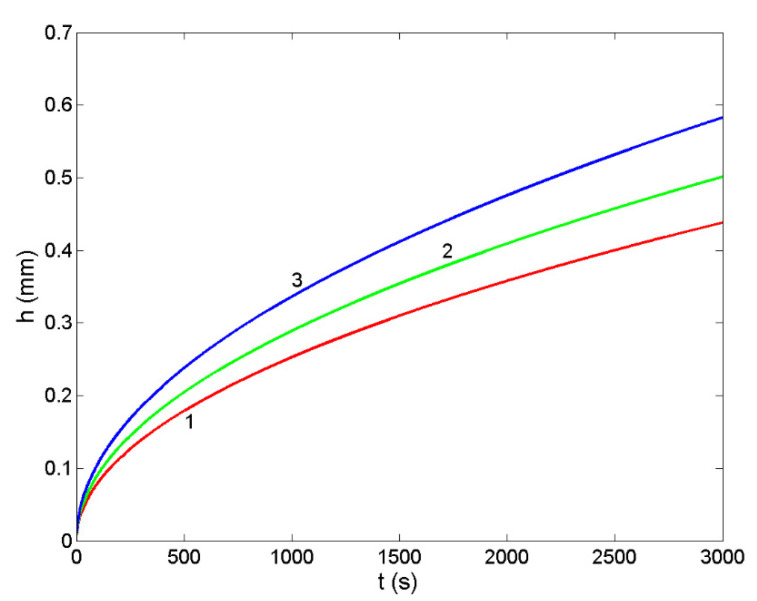
Infiltration depth of liquid Ir-80.5Si (at %) eutectic alloy into a porous SiC-substrate with pore sizes (in μm) of r = 65 (curve 1), r = 85 (curve 2) and r = 1.15 (curve 3), calculated for T = 1873 K by Washburn’s equation.

**Figure 10 materials-14-06024-f010:**
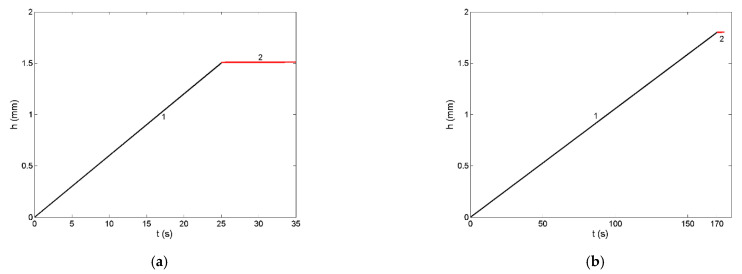
Infiltration curves for liquid $IrSi_{3}$/SiC system for (**a**) short- and (**b**) long-term experiments performed at T = 1523 and T = 1623 K [12]: the reactive infiltration (curves 1) and “similar to non-reactive” infiltration (curves 2).

**Table 1 materials-14-06024-t001:** Calculated values of the excess entropy of mixing ($\Delta S_{M}^{xs}$) and the excess energy ($\Delta G_{M}^{xs}$) of liquid Ir-Si alloys in the framework of the free volume theory. The enthalpy of mixing ($\Delta H_{M}$) and the excess volume ($\Delta V^{xs}$) values were calculated for T = 1873 K by Miedema’s model. (The meaning of symbols reported here is given in the Abbreviations).

*x_Si_*	Δ*H_M_* (kJ·mol^−1^)	$\Omega_{Si-Ir}$ (kJ·mol^−1^)	*P*	$U_{Ir}$ (kJ·mol^−1^)	$U_{Si}$ (kJ·mol^−1^)	$\Delta V^{xs}$ (cm^3^·mol^−1^)	$L_{Ir}$ (10^−8^ cm)	$L_{Si}$ (10^−8^ cm)
0.1	−9.202	−92.047	0.8006	−457.83	−407.02	−0.1877	1.414	1.454
0.2	−16.461	−82.441	0.6027	−447.52	−397.11	−0.3457	1.420	1.454
0.3	−21.739	−72.960	0.4096	−431.75	−386.51	−0.4970	1.428	1.454
0.4	−24.998	−64.340	0.2354	−409.27	−373.28	−0.6387	1.438	1.455
0.5	−26.200	−60.026	0.1455	−386.40	−348.51	−0.7665	1.448	1.459
0.6	−25.306	−65.053	0.2339	−382.42	−304.80	−0.8727	1.453	1.469
0.7	−22.277	−74.711	0.4086	−390.22	−261.29	−0.9425	1.454	1.478
0.8	−17.075	−85.493	0.6022	−400.64	−224.52	−0.9440	1.454	1.485
0.9	−9.662	−96.648	0.8.005	−411.72	−193.25	−0.7921	1.454	1.491
$x_{Si}$	$L_{Ir-Si}$ (10^−8^ cm)	$\Delta S_{VIB}^{xs}$ (J·K^−1^·mol^−1^)	$\Delta S_{CONF}^{xs}$ (J·K^−1^·mol^−1^)	$\Delta S_{M}^{xs}$ (J·K^−1^·mol^−1^)	$\Delta G_{M}^{xs}$ (kJ·mol^−1^)	$U_{IrIr}$ (kJ·mol^−1^)	$U_{SiSi}$ (kJ·mol^−1^)	
0.1	1.4542	−0.9548	−0.0906	−1.0454	−7.2435	−464.16	−165.95	
0.2	1.4541	−1.7936	−0.4027	−2.1963	−12.3472			
0.3	1.4540	−2.5055	−1.0079	−3.5133	−15.1588	$L_{IrIr}$ (10^−8^ cm)	$L_{SiSi}$ (10^−8^ cm)	
0.4	1.4538	−3.0713	−1.9258	−4.9971	−15.6388			
0.5	1.4538	−3.4509	−2.5974	−6.0484	−14.8714	1.4091	1.4955	
0.6	1.4538	−3.5526	−1.9418	−5.4945	−15.0145			
0.7	1.4540	−3.2887	−1.0163	−4.3049	−14.2137	$V_{Ir}$ (cm^3^·mol^−1^)	$V_{Si}$ (cm^3^·mol^−1^)	
0.8	1.4541	−2.6310	−0.4054	−3.0364	−11.3881			
0.9	1.4541	−1.5506	−0.0911	−1.6417	−6.5876	9.531	11.395	

## Data Availability

All data can be found within the manuscript.

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
