# Peer review of "Design of Composites by Infiltration Process: A Case Study of Liquid Ir-Si Alloy/SiC Systems"

_materials, 2021, doi:10.3390/ma14206024_

Round 1

Reviewer 1 Report

The manuscript by Novakovic et al. reviews the existent information of various melts such as IrSi and CoSi and their thermodynamics (Miedema’s model and free volume theory), density/molar volume of IrSi melts and following the theory structural information, surface segregation and surface tension, viscosity and non-reactive infiltration together with experimental data are specifically discussed for the case of the IrSi alloy /SiC system. The manuscript is quite comprehensive and with a well established discussion. Some points that can be improved include:

  1. Figure 3: the molar volume values of Co-Si melts from ref 60 is measured at 1773K while the concentration dependence of the molar volume of Co Si alloys. Despite the good, detailed explanation the authors have included in the text for the lower temperature data, it would be helpful if they could include the concentration dependance of the molar volume of liquid Co-Si alloys for 1773K too ( I believe it would be easier than finding some experimental data for the molar volume values of Co-Si melts at 1873K).
  2. Authors could refer in the Introduction that additional background related to theory (thermodynamic properties of metallic melts, structural information, etc.) will be further discussed in the Theory section. This as the discussed theory background is a crucial part for understanding the present work and needs to be better emphasized on.
  3. Some other minor points are related to improvements in the quality of figures: a)Text on the both X and Y is difficult to read, similar for the line annotations – this can be checked and improved for all figures b) Figure 7 – please check figure caption: “b“ ?, symbol of the dataset from ref 68

Author Response

Referee 1

Thank you very much for your comments and suggestions.

  1. Figure 3: the molar volume values of Co-Si melts from ref 60 is measured at 1773K while the concentration dependence of the molar volume of Co Si alloys. Despite the good, detailed explanation the authors have included in the text for the lower temperature data, it would be helpful if they could include the concentration dependance of the molar volume of liquid Co-Si alloys for 1773K too ( I believe it would be easier than finding some experimental data for the molar volume values of Co-Si melts at 1873K).

1) Using the experimental data of density of pure components obtained at T=1773 K [60], the molar volume isotherm of Co-Si melts was calculated and plotted on Fig. 3.

PAGE 8 LINE 8 (from the bottom)

“Therefore, for a comparison, the corresponding isotherm of Co-Si melts (Figure 3; curve 2) obtained at the same temperature together with the molar volume data deduced from the experimental data of density measured at T = 1773 K [60] are shown (Figure 3; curve 2). The dataset [60] has been obtained at lower temperature and therefore, the data values are lower with respect to the two isotherms. “

The text is replaced by

“Thus, for a comparison, the molar volume isotherm of Co-Si melts (Figure 3; curve 2) obtained at the same temperature together with the molar volume data deduced from the experimental data of density measured at T = 1773 K [60] including the corresponding isotherm (Figure 3; curve 3) are shown. The dataset [60] has been obtained at lower temperature and therefore, the experimental and theoretical data are lower with respect to the two isotherms calculated for T = 1873 K.“

Accordingly, the caption of Figure 3 was changed:

Figure 3. Concentration dependence of the molar volume of liquid Ir-Si (curve 1) and Co-Si (curve 2) alloys calculated for T = 1873 K. For a comparison, the molar volume of Co-Si melts (curve 3) calculated for T = 1773 K together with experimental values (˜) obtained from the density experimental data [60] measured at the same temperature are shown.

  1. Authors could refer in the Introduction that additional background related to theory (thermodynamic properties of metallic melts, structural information, etc.) will be further discussed in the Theory section. This as the discussed theory background is a crucial part for understanding the present work and needs to be better emphasized on.

2) The Introduction and Theory are rather long parts of the Manuscript, and to make them longer it does not make sense. Because of this, the authors added very short text in the Introduction, as follows:

PAGE 2 LINE 15 (from the bottom)

“Similar behaviour of the Co-Si and Ir-Si systems in terms of the mixing properties and the corresponding phase diagrams indicates that the thermophysical and structural properties of liquid Ir-Si alloys are very close to those of the Co-Si, described in detail in [41].”

  1. Some other minor points are related to improvements in the quality of figures: a)Text on the both X and Y is difficult to read, similar for the line annotations – this can be checked and improved for all figures b) Figure 7 – please check figure caption: “b“ ?, symbol of the dataset from ref 68

3a) Following Referee1’s comments, all Figures were redrawn using bigger size of fonts for X and Y labels as well as bigger size of symbols for literature data [60] and [68].

3b) Caption of Figure 7 was changed replacing blue circles by blue squares, as follows,

Figure 7. b. Viscosity isotherms of liquid Ir-Si (curve 1) and Co-Si (curve 2) alloys calculated by Moelwyn-Hughes’s (MH) model for T = 1773 K. For a comparison, the Co-Si viscosity isotherm (curve 2) and datasets (n) [68] are shown.

Reviewer 2 Report

In this article, based on the Washburn equation, a model of the percolation of a viscous mixture into a porous composite is constructed. In this case, the dependence of the viscosity of the mixture on the enthalpy of mixing is taken into account. In this paper, a sequential model of the seepage of a Sir mixture into a porous SiC is constructed, starting with the mixing enthalpy model for Si-IR. The comparison between different theoretical models is carried out and the qualitative coincidence of the model with experimental data is shown. The presented method of describing seepage is of interest to a wide range of researchers. Therefore, the article can be published without changes.

Author Response

Thank you very much to read and analysed our manuscript.

Reviewer 3 Report

Manuscripts designed to combine modeling with experiment are extremely valuable. The problem is and it is always difficult to check whether the proposed method will be applicable. It should also be remembered that the methods based on modeling have significant limitations and are not able to take into account all experimental factors. Recommend the work for publication without further corrections.

Author Response

(The authors gave the same response as above.)
